# MediYoga compared to physiotherapy treatment as usual for patients with stress-related symptoms in primary care rehabilitation: A randomized controlled trial

Madeleine Bellfjord[1], Anna Grimby-Ekman[2], Maria E. H. Larsson[3,4]*

**1** Region Västra Götaland, Department of Rehabilitation, Närhälsan Clinic in Primary Care, Gibraltarg, Gothenburg, Sweden, **2** School of Public Health and Community Medicine, Institute of Medicine, Sahlgrenska Academy, University of Gothenburg, Gothenburg, Sweden, **3** Research, Education, Development and Innovation Primary Health Care, Region Västra Götaland, Gothenburg, Sweden, **4** Department of Health and Rehabilitation, Institute of Neuroscience and Physiology, Sahlgrenska Academy, University of Gothenburg, Gothenburg, Sweden

* maria.eh.larsson@vgregion.se

**Data Availability Statement:** All relevant data are within the paper and its Supporting Information files.

## Abstract

### Objective

The purpose of this study was to compare the effect of MediYoga as a group treatment to conventional treatment provided by a physiotherapist for people with perceived stress-related symptoms.

### Design

Randomized controlled trial.

### Settings

Primary care rehabilitation, Gothenburg Sweden.

### Subjects

Fifty-five patients with stress-related symptoms were invited to participate. Nine patients declined, and a total of 46 patients aged 26–70 years (mean 47), 44 women and two men were randomized, 23 to the MediYoga group and 23 to the physiotherapy treatment as usual group.

### Interventions

The MediYoga group performed MediYoga for one hour a week during an 8-week period. The control group received physiotherapy treatment as usual.

### Main measures

Data were mainly collected by self-reported questionnaires. For primary outcome the Swedish version of the Perceived Stress Scale (PSS) was used. Secondary outcomes were the

**Funding:** MB VGFOUGSB-602221 The Local Research and Development Council Gothenburg and Södra Bohuslän. The funders had no role in study design, data collection and analysis, decision to publish, or preparation of the manuscript.

**Competing interests:** The authors have declared that no competing interests exist.

Hospital Anxiety and Depression Scale (HADS), EuroQol–5D (EQ-5D) and EuroQol–Visual Analog Scale (EQ-VAS). Thoracic excursion was the only physical measurement. Mixed effect model was used for analyse.

## Results

For the primary outcome PSS, there was a close to statistically significant group effect over time advantaging MediYoga over physiotherapy ($P = 0.06$). For secondary outcomes, the group effect over time was statistically significant in HADS anxiety ($P = 0.01$) and EQ-VAS ($P = 0.03$). There was a group trend over time advantaging MediYoga in HADS depression ($P = 0.08$).

## Conclusion

Despite a large dropout in both groups, MediYoga can be recommended as a treatment option for people suffering from stress-related symptoms.

## Trial registration

**Registered in:** ClinicalTrials.gov NCT02907138

## Introduction

Society and healthcare face major challenges with the increase of reported stress-related disorders [1]. Stress as a phenomenon is difficult to define and has been controversial since it was first described by Selye in 1936 [2]. Selye was a pioneer in describing the concept of stress, which he suggested was ingrained in the vocabulary of daily life; he also described underlying mechanisms and hypothesized the existence of a "first mediator" in the hypothalamus [3]. A new concept of the cycle of stress reaction was presented in 2015 [4] trying to bridge the gap between physiology and psychology. The cycle describes a circular event in life, composed of four phases: the resting ground phase, the tension phase, the response phase and the relief phase. Stress as a concept is not a disease but is often described as stress linked to exposure to and/or the experience of stress.

Different treatments have been described to reduce stress and anxiety, such as multimodal intervention (MMI) and cognitive behavioural therapy (CBT) [5]. Results from a randomized controlled study in a primary care setting showed significant results for the intervention with MMI compared to CBT and care as usual (CAU). CBT showed significance results over CAU for people with stress-related disorders [5], CBT and anti-depression medication1 should be offered by healthcare provision for mild to moderate depression and anxiety, as recommended treatment. In addition, for anxiety, a supplemental therapy could be offered, such as physical activity as well as body awareness therapy and mindfulness-based stress reduction.? Nonetheless, there is still a scarcity of international and national guidelines for the best practice for stress-related disorders.

Increasing interest is shown for mind–body techniques such as yoga. Yoga has shown to decrease stress levels by restoring the body's sympathetic–parasympathetic balance [6]. The relationship between yoga and stress have been discussed within a theoretical perspective that refers to psychology as embodiment, parallels the connection between mind and body and suggests that yoga may influence stress by influencing the musculoskeletal system [7]. There is

a gap in knowledge, though, between how yoga affects stress in the musculoskeletal system, and psychological variables and processes, and the interactions between those variables [7]. During the last decade several different alignments of yoga have been evaluated according to the effect of yoga on the levels of stress and well-being. The effect of different yoga programs on perceived stress and psychological outcomes has been investigated in randomized controlled trials (RCTs) in different populations, such as distressed women; [8] physically inactive, stressed adults; [9] and graduate students and staff at a university faculty [10]. Results from these studies showed significant improvements in perceived stress and symptoms of stress in multiple realms of functioning–from psychological to behavioural8–10 as well as for general health-related quality of life (HRQoL) [8–10]. In a recently (2021) published RCT, [11] nursing staff suffering from various levels of stress were randomized to an intervention group of 12 weeks of structured yoga or a waitlisted control group. At follow up, the Perceived Stress Scale (PSS) score differed significantly in favour of the intervention group compared to the waitlisted control group.

MediYoga (MY) is a mind–body intervention including specific mind–body techniques, founded in 1997 in Sweden [12]. It was developed as a therapeutic, structured method available to all patients, regardless of different kinds of illnesses. MY is commonly performed lying or sitting on the floor, but can also be performed sitting on a chair. MY [12] is the only form of all alignments of yoga that is accepted and approved in Swedish health and medical care. MY is today offered in approximately 300 clinics in health and medical care all over Sweden and in a number of rehabilitation units within Region Västra Götaland (VGR). MY was considered to be cost-effective and increased quality of life more than standard treatment (physiotherapy, PT) in patients with low-back pain [13]. In two studies from 2017 and 2020, [14, 15] MY showed improved HRQoL and decreased blood pressure in patients with paroxysmal atrial fibrillation compared to control groups. In a study from 2013, MY was compared to standard care for patients with stress-related symptoms and diagnoses in primary healthcare. Significant results were found regarding self-estimated stress, anxiety, HRQoL and thoracic excursion after performing MY [16]. In recent years there has been a noticeable increase in numbers of people with stress and stress- related symptoms seeking primary care rehabilitation. Generally, PT is offered as the first treatment option for these patients. However, it is a challenge to meet the demand, and there is a need to evaluate more specific treatment methods for people suffering from stress and anxiety.

The healthcare system is still inadequately equipped to meet the needs of interventions for less severe forms of stress and stress-related symptoms; only guidelines for anxiety and depression are available [17]. To the best of our knowledge, no previous study has compared MY to PT only– treatment as usual–for people suffering from stress-related symptoms.

The purpose of this study was to compare the effect of MY as a group treatment to conventional treatment provided by a physiotherapist, for people with perceived stress-related symptoms.

## Methods

The study was a two-armed RCT. The patients were recruited through advertising in a local newspaper or by physiotherapists who worked at the Primary Care Rehabilitation clinic. All patients were informed both orally and in writing about the study and gave both verbal and written consent to participate in the study. The inclusion criteria were self-reported symptoms from stress, and also stress-related symptoms like anxiety, musculoskeletal discomfort from their neck and shoulders, headache and sleeping problems.

Exclusion criteria were the inability to understand the Swedish language, pregnancy, psychotic symptoms, participation in PT currently or during the past three months or any previous participation at all in MY, either group or individual. Other medical or psychological treatment was maintained during the intervention period. A total of 55 individuals were invited to participate in the study and attended Primary Care Rehabilitation, Gothenburg, Sweden. Nine participants declined at the time for invitation. A block-randomized generated list with four blocks with ten in each block, produced from a computer-generated list, was used for the randomization. Group allocation was prepared in sequentially numbered, sealed black envelopes, by an independent person. When the patients agreed to participate in the study, randomization was conducted by an independent person in a telephone conversation with the patient. Recruitment started in September 2016 and ended in October 2018. The study was approved by the Regional Ethical Review Board in Gothenburg, Gothenburg, Sweden (Dnr:658–16). The study is registered in - ClinicalTrials.gov NCT02907138.

A total of 46 patients were randomized for the study, 23 to the intervention group (MY), and 23 to the physiotherapy group treatment as usual (PT). Baseline characteristics of patients included in MediYoga and physiotherapy groups, presented in Table 1.

## MediYoga group

The patients attended the MY group once a week over an 8-week period. The class lasted for 60 minutes and was led by a physiotherapist who also is a certified MY teacher and therapist (first author of the article). During the first four weeks, patients were guided in and practiced the MY Start-Up package 1 (Appendix 1 in S2 File) and during the four remaining weeks the Start-Up package 2 (Appendix 2 in S3 File). The Start-Up package 1 and 2 always started with MY breathing in a lying position (about 10–15 minutes). Guided gentle exercises were then performed while sitting in an easy meditation position or while sitting on a chair. Mantras were also included as a natural part of Start-Up package 1 and 2. Each exercise was performed for about 3 minutes, during a session of about 20–25 minutes. This was followed by a long relaxation (Savasana) for 10–15 minutes, and the session ended with 5–10 minutes of meditation, according to Start-Up package 1 and 2. The exercises in Start-Up package 1 and 2 were performed according to the MediYoga of Sweden Institute [12].

## Physiotherapy group

Patients who received physiotherapy treatment as usual were offered physiotherapy in an eight-week period. The number of visits to a physiotherapist could vary from one visit for assessment or one to two visits a week for treatment. Physiotherapy treatment as usual, could

**Table 1. Baseline characteristics of patients included in MediYoga and physiotherapy group and the average participation in treatment sessions.**

| Recruitment period 2016–2018 | N | Age years Mean; median (SD; range) | Sex F/M | Participation during 8 weeks, session in mean (range) | Participation during 8 weeks, sessions in median |
|---|---|---|---|---|---|
| Mediyoga and Physiotherapy group | 46 | 47;47 (12.1; 26–70) | 44/2 | | |
| MediYoga (randomized) | 23 | 48;47 (12.8; 28–70) | 23/0 | 3.5 (0–8) | 4 |
| Physiotherapy (randomized) | 23 | 46;47 (12.8; 26–65) | 21/2 | 4.4 (0–18) | 4 |

for example be physical exercise or body awareness therapy. The treatment time for each occasion was between 30 and 45 minutes.

## Outcome measures

Baseline measurements in all patients were performed one week before the intervention period started. Measurements were repeated in both groups 8 and 20 weeks after the start of intervention. The data collection was carried out at the rehabilitation clinic in primary care, and at follow-up, the surveys including a pre-paid return envelope, were sent out to all the participants. The surveys could not be delivered by the web-based system as planned due to fact that the web-based system was not sufficiently developed to carry out a questionnaire mailing to the study participants. To measure the excursion of the chest, patients needed to attend the rehab clinic, even at follow-up after 8 and 20 weeks.

## Primary outcome

To measure the general stress level, the Swedish version of the PSS [18] was used. The instrument was developed by Cohen et al. and translated into Swedish by Eskin, Parr and Eklund [19]. The instrument includes 14 questions on perceived stress in the previous month. Responses are given on a scale ranging from 0 (never) to 4 (very often). The total score is calculated with a possible score range of 0–56. A higher score reflects a higher degree of perceived stress. There are no limits for categories, light or moderate or severe stress, in this scale. The PSS is used to compare before and after treatment between groups or individuals. Eklund et al. [19] showed PSS to be feasible with the investigated samples, and the results indicated no ceiling or floor effects and good internal consistency of the PSS. Several aspects of construct validity were shown. The Swedish version of the PSS showed satisfactory psychometric properties in coping, homogeneity, reliability and validity. Their conclusions were that the Swedish version of PSS may be recommended for use in both people with and without known stress-related disorders [19].

## Secondary outcomes

To assess symptoms of anxiety and depression, The Hospital Anxiety and Depression Scale (HADS) [20], was used. The scale consists of 14 items, 7 items for anxiety and 7 items for depression, measuring symptoms on a Likert scale with a score range of 0–3, a higher rating indicating a higher state of anxiety or depression symptoms. The total score is calculated with a possible range of 0–21 for the 7 items of anxiety and the 7 items of depression. The limit for moderate symptoms are 8–10 points, and the range over 11 points indicates that anxiety and depression symptoms are present. Zigmond et al. [20] report that the self-assessment scale has been developed and found to be a reliable instrument for detecting states of depression and anxiety in the setting of an hospital medical outpatient clinic. The anxiety and depression sub-scales are also valid measures of severity of the emotional disorder [21]. The structure of HADS was evaluated in a Swedish population sample, and the results indicated that the factor structure was quite strong, consistently showing two factors in the whole sample as well as in different subsamples [16].

To describe and measure HRQoL, EuroQol–5D [22] (EQ-5D) was used. It consists of a questionnaire where the individual rates self-assessed health in five dimensions (mobility, self-care, usual activities, pain/discomfort, anxiety/depression) in three degrees, no problems = 1, moderate problems = 2, severe problems = 3. An EQ-5D index summary is derived by applying a formula that essentially attaches values (weights) to each of the levels in each dimension [20]. In the current study the Swedish version was used, EuroQoL Group 1990. In EQ-5D

there is also a EuroQol–Visual Analogue Scale (EQ-VAS) included, which is used as a quantitative measure of overall health status. The range is presented in a vertical line on a 0–100 hash-marked, vertical visual analogue scale (EQ-VAS) where 0 represents the worst and 100 the best health state the respondent can imagine. The patients marked with a cross on the line to describe their perception of their overall health status. The EQ-5D has been widely tested and used in both general population and patient samples and has been translated into over 130 different language versions [www.euroqol.org].

Thoracic excursion was the only physical measurement. A measuring tape marked in millimeters was used to measure the extent of chest expansion or mobility. Thoracic excursion was measured at the level of the xiphoid process, performed with the patient standing, hands placed on head, and given the instruction to "breathe into the fullest extent and make yourself as big as possible, then breathe out and make yourself as small as possible". The procedure has been described in the study by Olsén et al. [23]. The difference between inhalation and exhalation was measured in millimeters, and the procedure was repeated two times, with the largest difference being registered. The measurement has been shown in previous studies [16, 23] capable of detecting a statistically significant difference.

## Statistical analysis

All data were analysed using SPSS version 25.0 (SPSS Inc., Chicago, IL, USA). A t-test was used when power was calculated for the primary outcome general stress level, measured with the PSS. Including 16 patients in each group would yield 80% of power (alpha = 0.05) to detect a difference of 7 units between the groups, assuming a standard deviation (SD) of 7.0, calculating from SD as described by Brinkborg et al [24]. Since dropouts were anticipated to be 15%–20%, we wanted to include another four patients in each group so that the total number of patients in the study would be 40. All measurements were performed at baseline and after 8 and 20 weeks after the treatment had started. For the missing analysis of dropouts, two-sided t-test was used ($P < 0.05$).

Treatment effects, regarding both the primary outcome and the secondary outcomes, were analyzed using mixed-effects linear models with a random intercept. The covariance structure used in the mixed effects model is the compound symmetry for the random effects. This was to account for our data consisting of repeated measures on individuals over time, and hence, a dependency structure between observations being present [25]. Age was considered a potential confounder. The criteria for inclusion of the confounder was if $P < 0.2$, or if $P > 0.2$ and more than 15% changed in the estimate of the main fixed effects, not the estimates for the confounders. These criteria for inclusion were not met for any of the outcomes; hence, the models were not adjusted for age.

## Results

A total number of 46 patients, were included in the study. Of the 46 patients included (44 women, and 2 men), there were 13 patients remaining in the MY group and 10 patients in the PT group at 8 weeks' follow-up, and at 20 weeks' follow-up there were still 10 patients remaining in each group, see Fig 1.

The number of patients who received physiotherapy treatment as usual are described in four categories after analyzing the treatments of physiotherapy. This analyze was performed of an independent physiotherapist with permission from operations manager after the study was clinically closed. From the patients journal the physiotherapist could read what type of physiotherapy treatment the patient had received.

## Flowchart

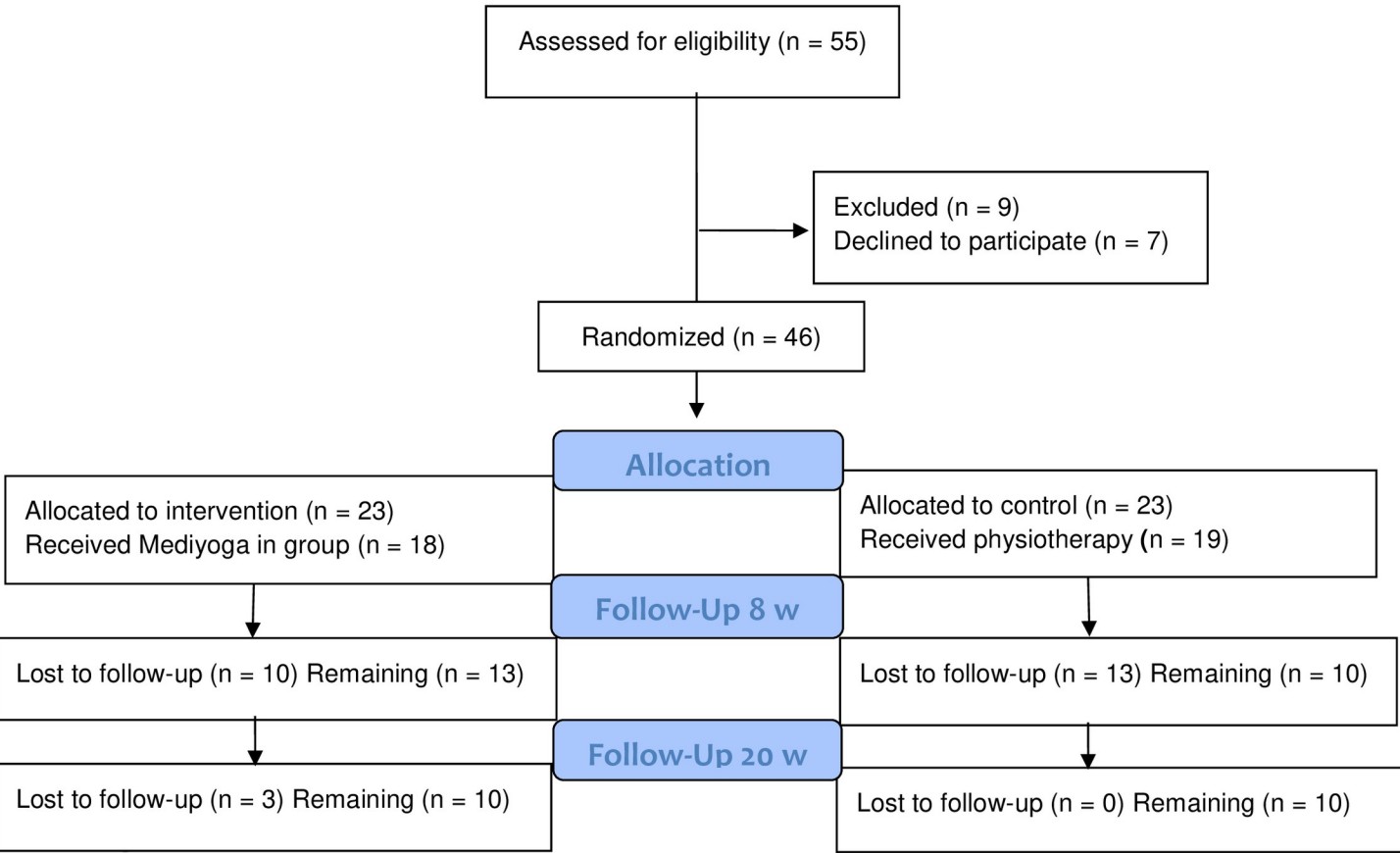

**Fig 1. Flowchart over time, MediYoga versus physiotherapy.**

Physiotherapy treatment as usual are described in four categories 1) body and mind, 2) exercise therapy, 3) treatment of pain and 4) assessment of sole treatment, presented in Table 2.

The number of patients in the physiotherapy group who received treatment in one of the four categories or treatment combination from two categories, presented in Table 3.

**Table 2. Physiotherapy treatment options categorized based on treatment modality.**

| Body and mind | Exercise Therapy | Treatment of pain | Assessment of sole treatment |
|---|---|---|---|
| Body awereness therapy | Medical exercise therapy | Acupuncture | Assessment |
| Relaxation breathing exercise | Information about physical activity | Information about pain | |
| Stress management | Home exercise program | | |
| Sleep hygiene | | | |
| Mindfulness | | | |

**Table 3. Number of patients in the physiotherapy group who received treatment in one of the four categories or treatment combination from two categories.**

| Physiotherapy treatment, in categories | N 23 | Percentage 100 |
|---|---|---|
| Body and mind | 4 | 17 |
| Exercise therapy | 4 | 17 |
| Treatment of pain | 0 | 0 |
| Only Assessment | 2 | 9 |
| Combination Body and mind and exercise therapy | 8 | 35 |
| Combination Exercise therapy and Treatment of pain | 2 | 9 |
| No assessment or treatment | 3 | 13 |

The results from the mixed-effects linear regression for the primary outcome PSS showed a different time pattern between MY and PT; the interaction between time and group was close to statistically significant ($P = 0.06$) (Tables 4 and 5). The MY group had an advantage in the decrease of PSS. Model means from the mixed-effects linear regression model, including group, time and group–time Interaction, see Fig 2.

For the secondary outcomes the interaction terms, was statistically significant for HADS anxiety ($P = 0.01$) and EQ-VAS ($P = 0.03$). The eight-week result showed that the MY group decreased in anxiety to a larger extent than the PT group, even the level of anxiety was similar at 20 weeks. For EQ5D the MY group increased to a larger extent than the PT group both at eight and twenty weeks, Table 5. For HADS depression there was a trend for the interaction ($P = 0.08$); see Tables 4 and 5.

## Analysis of dropouts from baseline

At 8 and 20 weeks' follow-up in both the MY and PT groups there was a large dropout rate. Baseline values for age, PSS, HADS anxiety, HADS depression, EQ- 5D and EQ-VAS were compared between the group of those who answered the questionnaire at 8 weeks (n = 20) and those who had dropped out at 8 weeks (n = 17). For HADS depression there was a significant difference ($P = 0.03$) between the group of those who answered the questionnaire at 8 weeks (n = 20) and those who had dropped out at 8 weeks (n = 17). In the dropout group the baseline mean on the HADS depression scale was higher than for the non-dropouts.

## Discussion

The major findings were that for MY as a group treatment intervention for 8 weeks, showed reduced levels of stress and anxiety and increased HRQoL compared to PT treatment as usual."

To the best of our knowledge, this is the first study to evaluate the effect of MY in patients with stress-related symptoms compared to solely PT treatment as usual in primary care rehabilitation. The results are in concordance with previous studies including yoga, [8–10], [16, 26] with reduced stress and anxiety and increased HRQoL. In the PT group 12 of 23 patients had PT treatment in the category body and mind. As treatment in this category has similar impact in the autonomic nervous system as do the MY treatment, this may explain why the differences were not so great between the MY group and the PT group regarding the experience of stress and depression.

In the current study results from the outcome thoracic excursion were not statistically significant between the MY group and the PT group, though the model showed tendencies of

**Table 4. Parameter estimates, standard errors and *P*-values for the fixed effects, in the mixed-effects regression models for each outcome.**

| | | | Parameter estimate | SE[*] | P-value |
|---|---|---|---|---|---|
| **PSS** | **intercept** | | 29.8 | 1.98 | <0.001 |
| | **Group** (ref physical therapy) | MediYoga | -3.0 | 2.80 | 0.228 |
| | **Time** (ref Follow up 20 weeks) | Baseline | 2.5 | 2.52 | 0.001 |
| | | Follow Up 8 weeks | -0.9 | 2.80 | |
| | **Group*Time** (ref physical therapy* Follow up 20 weeks) | MediYoga* Baseline | 5.7 | 3.48 | 0.060 |
| | | MediYoga* Follow Up 8 weeks | -3.0 | 3.84 | |
| HAD Anxiey | **intercept** | | 8.8 | 1.26 | <0.001 |
| | **Group** (ref physical therapy) | MediYoga | -0.1 | 1.64 | 0.912 |
| | **Time** (ref Follow up 20 weeks) | Baseline | 2.1 | 1.36 | <0.001 |
| | | Follow Up 8 weeks | 0.5 | 1.24 | |
| | **Group*Time** (ref physical therapy* Follow up 20 weeks) | MediYoga* Baseline | 2.0 | 1.77 | **0.014** |
| | | MediYoga* Follow Up 8 weeks | -1.9 | 1.61 | |
| HAD Dep | **intercept** | | 5.9 | 1.19 | <0.001 |
| | **Group** (ref physical therapy) | MediYoga | -1.6 | 1.57 | 0.438 |
| | **Time** (ref Follow up 20 weeks) | Baseline | 1.4 | 1.08 | 0.001 |
| | | Follow Up 8 weeks | 0.2 | 0.97 | |
| | **Group*Time** (ref physical therapy* Follow up 20 weeks) | MediYoga* Baseline | 2.3 | 1.40 | 0.076 |
| | | MediYoga* Follow Up 8 weeks | -0.4 | 1.27 | |
| EQVAS | **intercept** | | 53.4 | 7.00 | <0.001 |
| | **Group** (ref physical therapy) | MediYoga | 16.2 | 9.19 | 0.392 |
| | **Time** (ref Follow up 20 weeks) | Baseline | 3.6 | 7.03 | 0.002 |
| | | Follow Up 8 weeks | 8.0 | 6.28 | |
| | **Group*Time** (ref physical therapy* Follow up 20 weeks) | MediYoga* Baseline | -23.8 | 9.10 | **0.027** |
| | | MediYoga* Follow Up 8 weeks | -10.9 | 8.19 | |
| EQ5D | **intercept** | | 0.69 | 0.051 | <0.001 |
| | **Group** (ref physical therapy) | MediYoga | 0.13 | 0.067 | 0.166 |
| | **Time** (ref Follow up 20 weeks) | Baseline | -0.06 | 0.060 | 0.007 |
| | | Follow Up 8 weeks | -0.03 | 0.047 | |

(*Continued*)

**Table 4.** (Continued)

|  |  |  | Parameter estimate | SE[*] | P-value |
|---|---|---|---|---|---|
|  | **Group*Time** (ref physical therapy* Follow up 20 weeks) | MediYoga* Baseline | -0.16 | 0.080 | 0.148 |
|  |  | MediYoga* Follow Up 8 weeks | -0.07 | 0.061 |  |
| Thoracic excursion | **intercept** |  | 5.6 | 0.73 | <0.001 |
|  | **Group** (ref physical therapy) | MediYoga | 1.8 | 1.02 | 0.256 |
|  | **Time** (ref Follow up 20 weeks) | Baseline | -0.01 | 0.591 | 0.133 |
|  |  | Follow Up 8 weeks | 0.7 | 0.60 |  |
|  | **Group*Time** (ref physical therapy* Follow up 20 weeks) | MediYoga* Baseline | -1.3 | 0.80 | 0.203 |
|  |  | MediYoga* Follow Up 8 weeks | -1.3 | 0.80 |  |

SE = standard error

improved thoracic excursion over time. This is contrary to two previous studies [16, 23] that showed significant results for thoracic excursion, also measured with a tape. A possible factor in the lack of significant results could be the circumstances of having two different physiotherapists measuring thoracic excursion. This was not the plan at the outset, but the physiotherapist who started measuring the thoracic excursion went on maternity leave and another physiotherapist continued the measuring, so that the study could be completed. For further research and evaluations of thoracic excursion, an inter-assessor reliability test needs to be performed before starting to measure thoracic excursion.

There was no statistically significant difference between the MY and PT groups regarding the outcome HADS depression. This result is consistent with previous research as in a randomized controlled trial [16] where MY was allocated for one hour once a week over a 12-week period, in addition to the standard care." It may be that a longer treatment period with MY is required, more than 8 weeks according to the current study and even more than 12 weeks, [16] to be able to influence the degree of depression, as is also discussed by the authors of the 12-week study [16]. This has also been suggested by Javnbakht et al [27]. In that study, yoga was the intervention compared to a waiting list. The results showed no significant differences for the outcome HADS depression, although the participants were offered yoga classes of 90 minutes two times a week for two months. The matter of frequency and intensity was

**Table 5. Estimated model means from the linear mixed effects model.**

|  | Baseline |  |  |  | Follow up 8 w |  |  |  | Follow up 20 w |  |  |  |
|---|---|---|---|---|---|---|---|---|---|---|---|---|
|  | Medi Yoga |  | Physio Therapy |  | Medi Yoga |  | Physio Therapy |  | Medi Yoga |  | Physio Therapy |  |
|  | mean | 95% CI | mean | 95% CI | mean | 95% CI | mean | CI 95% | mean | CI 95% | mean | CI 95% |
| HAD Anxiey | 12.7 | 11.2–14.2 | 10.8 | 9.3–12.4 | 7.3 | 6.0–8.6 | 9.3 | 7.7–10.9 | 8.7 | 6.4–10.8 | 8.8 | 6.2–11.4 |
| HAD Dep | 8.0 | 6.2–9.7 | 7.2 | 5.4–9.1 | 4.1 | 2.3–6.0 | 6.1 | 4.1–8.2 | 4.3 | 2.2–6.5 | 5.8 | 3.4–8.3 |
| EQVAS | 49.4 | 41.6 - 57.3 | 57.0 | 48.9–65.2 | 66.7 | 59.5–73.9 | 61.4 | 53.3–69.6 | 69.6 | 57.2–82.1 | 53.4 | 38.7–68.1 |
| EQ5D | 0.5 | 0.5–0.7 | 0.6 | 0.5–0.7 | 0.7 | 0.7–0.8 | 0.6 | 0.6–0.7 | 0.8 | 0.7–0.9 | 0,6 | 0.6–0.8 |
| Thoracic excursion | 6.0 | 4.8–7.4 | 5.5 | 4.3–6.7 | 6.8 | 5.4–8.3 | 6,2 | 4.9–7.7 | 7.4 | 6.0–8.9 | 5.5 | 4.1–7.1 |

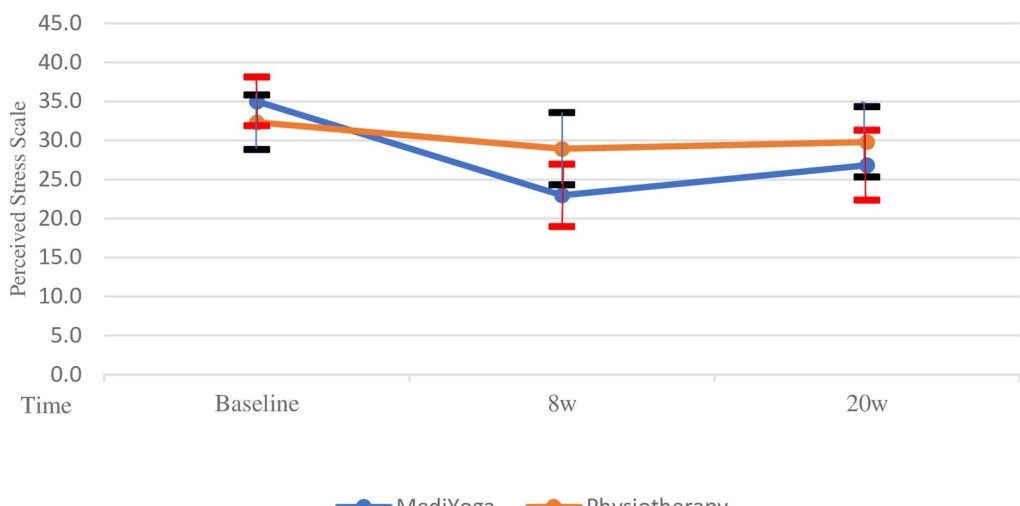

**Fig 2. Perceived Stress Scale. Groups of MediYoga and physiotherapy 95% confidence intervals are presented.**

shown in a meta-analysis by Brinsley et al., [28] where 13 studies were included with 632 participants aged 18 years or older with diagnosis of any mental disorder. The purpose was to see what effects yoga might have for people with depressive symptoms in mental disorders. Yoga interventions were on average 2.4 months long with 1.6 sessions per week (range 1–3 sessions) of on average 60 minutes' duration (20–90 minutes). Yoga showed greater reductions in depressive symptoms than waitlist treatment as usual and attention control. The conclusion was that greater reductions in depressive symptoms were associated with higher frequency of yoga sessions per week; [28] the intensity had no effect on depressive symptoms according to whether the yoga was offered for between 20 and 90 minutes [28].There is a need to elaborate on the questions of what frequency and intensity of MY are required to reduce depressive symptoms.

Strengths of the current study are that results showed significant results for anxiety and HRQoL advantage for MY, despite the fact that the participants only participated in half of the offered opportunities. In the future this is an aspect that can be taken into account in the number of hours offered for participation and implementation of MY groups. The treatments were also performed as part of regular activities in a primary care rehabilitation clinic and can easily be implemented. The major limitation in the current study was the large dropout rate. After 8 weeks of participation almost 50% in both groups had dropped out. A possible cause and resistance to complying with treatment could be that participants needed to go from home or their workplace to participate in the MY class or PT treatment. The dropout rate was comparable to a study by Mandal et al. [11] where nursing staff suffering from various levels of stress and burnout were randomized either to intervention, 12 weeks of a structured yoga group, or to a wait-list control group. Of the total enrolled, 45% of participants completed the study. The reasons for dropout cited by 55% of participants were issues of time and health issues of family members. In the meta-analysis by Brinsley et al.28 with 13 included RCTs to evaluate effects of yoga for people with depressive symptoms in mental disorders, the total dropout rate ranged from 9% to 48%.

Students and staff with academic stress at the universities seem to have better adherence to completing study participation. A study by Brems et al [26] recruited a total of 50 students and a group of faculty members with academic stress to participate in research evaluating yoga with the primary purpose to test the feasibility of bringing yoga into the academic workplace

as a regular and valued activity [26]. The intervention consisted of 10 weekly 90-minute sessions that were structured to include conceptual grounding, breathing, postures and meditation. Of 44 students and staff (40 women and four men), 88% completed the study. Adherence rates revealed that once individuals. were engaged in the practice they created the time in their busy schedules to attend classes at least 80% of the time. In another study by Prasad et al., [29] the aim was to determine the effect of six weeks of yoga and meditation on medical students' levels of perceived stress and sense of well-being. A total of 34 students were included, and 7 dropped out before intervention had started. The yoga sessions took place at the university campus.29 Seventy-nine per cent out of 27 students (13 women and 14 men) completed the study. Results showed a statistically significant reduction in perceived stress after the 6-week yoga and meditation program. The advantage is that the students and staff at universities were offered yoga classes directly after lessons at the university campus. This could be one factor explaining better compliance with participation compared to the current study. For someone suffering from stress, anxiety and depressive symptoms, it takes more effort to attend a treatment away from home or work compared to students who probably are more motivated to participate overall and have convenient access to treatment on site.

More research is needed to evaluate the optimal frequency, intensity and length of treatment period where adherence to treatment still remains. Most previous studies in yoga are evaluated with questionnaires for self-rated stress, anxiety, depression and HRQoL. In primary care rehabilitation, there is also a need to develop a validated tool that is easy to use for measuring physiological levels of stress and to explore whether any physiological changes in stress may occur from baseline to post intervention.

## Conclusion

MediYoga as a group treatment seems to have good effects on perceived stress, anxiety and health related quality of life compared to physiotherapy.

Although there was a large dropout rate in both groups, we can recommend MediYoga for people suffering from stress-related symptoms.

## Supporting information

**S1 Checklist.**
(DOCX)

**S1 Data.**
(XLS)

**S1 File.**
(DOCX)

**S2 File.**
(PDF)

**S3 File.**
(PDF)

**S4 File.**
(DOCX)

**S5 File.**
(DOCX)

**S6 File.**
(DOCX)

## Acknowledgments

The authors would like to thank the management and all the physiotherapists working between 2016–2018 in primary care rehabilitation Närhälsan Gibraltar in Gothenburg, who contributed to the realization of the study. Special thanks to physiotherapist Amanda Öhrn and Julia Sunnerholm who performed the physical measurements for all patients in the study and to rehabilitation assistant Marie Granath who had a significant role in the process of recruitment and randomization of study participants.

## Author Contributions

**Conceptualization:** Madeleine Bellfjord, Maria E. H. Larsson.

**Data curation:** Madeleine Bellfjord.

**Formal analysis:** Madeleine Bellfjord, Anna Grimby-Ekman, Maria E. H. Larsson.

**Funding acquisition:** Madeleine Bellfjord.

**Investigation:** Madeleine Bellfjord.

**Methodology:** Madeleine Bellfjord, Maria E. H. Larsson.

**Project administration:** Madeleine Bellfjord.

**Supervision:** Maria E. H. Larsson.

**Visualization:** Madeleine Bellfjord, Anna Grimby-Ekman, Maria E. H. Larsson.

**Writing – original draft:** Madeleine Bellfjord.

**Writing – review & editing:** Madeleine Bellfjord, Anna Grimby-Ekman, Maria E. H. Larsson.

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
