## [Decision Letter · Decision Letter 0]

22 Aug 2023

PONE-D-23-05385MediYoga compared to physiotherapy treatment as usual for patients with stress-related symptoms in primary care rehabilitation: a randomized controlled trialPLOS ONE

Dear Dr. Bellfjord,

Thank you for submitting your manuscript to PLOS ONE. After careful consideration, we feel that it has merit but does not fully meet PLOS ONE’s publication criteria as it currently stands. Therefore, we invite you to submit a revised version of the manuscript that addresses the points raised during the review process.

We look forward to receiving your revised manuscript.

Kind regards,

Enock Madalitso Chisati, PhD

Academic Editor

PLOS ONE

Journal Requirements:

Reviewers' comments:

Reviewer's Responses to Questions

**Comments to the Author**

1. Is the manuscript technically sound, and do the data support the conclusions?

Reviewer #1: Yes

Reviewer #2: No

Reviewer #3: Yes

2. Has the statistical analysis been performed appropriately and rigorously? 

Reviewer #1: Yes

Reviewer #2: Yes

Reviewer #3: Yes

3. Have the authors made all data underlying the findings in their manuscript fully available?

Reviewer #1: No

Reviewer #2: Yes

Reviewer #3: Yes

4. Is the manuscript presented in an intelligible fashion and written in standard English?

Reviewer #1: Yes

Reviewer #2: No

Reviewer #3: Yes

5. Review Comments to the Author

Reviewer #1: A two-arm randomized clinical trial was conducted which aimed to compare the effect of MediYoga to conventional treatment for people with perceived stress-related symptoms. For the primary outcome, Perceived Stress Scale, the MediYoga group showed an advantage over physiotherapy, but the difference did not reach statistical significance. As for secondary outcomes, the group effect over time was statistically significant in HADS anxiety and EQ-VAS.

Minor revisions:

1- In the abstract, briefly identify the statistical testing methods used to estimate the p-values.

2- Table 1: For age, provide the mean, standard deviation, and range. For sex, in addition to the frequencies, provide the corresponding percentages.

3- Table 3: In addition to the frequencies, provide the corresponding percentages.

4- Page 9: Statistical Power Calculation: State the statistical testing method which attains 80% power. Perhaps it is a t-test.

5- Page 9: The t-test is denoted with lower case t.

6- Page 9: Perhaps “t-test 2 signed” is a typo for “two-sided t-test.”

7- Page 9: Indicate the underlying covariance structure used in the mixed-effects linear models.

8- Table 5: Clarify if CI represents 95% CI.

9- Table 5: Merge the cells for 8 week follow-up and 20 week follow-up, separately, in the first row. Use the same cell merging format that has been applied to the Baseline section.

10- Clearly describe the results of significant interaction effects.

11- To assist in the review process, add line numbering to the document.

Reviewer #2: Summary

The study was conducted to examine the effects of MediYoga (MY), in comparison to Physiotherapy management techniques, on stress, anxiety and depression measurement scores among adults presenting with the associated symptoms such as musculoskeletal discomfort, headache and sleeping disorders. No clinical diagnoses were made on such conditions in this study; participants self-reported the symptoms using the self-administered questionnaires. Measures of stress constituted the primary outcome, whereas the anxiety and depression measures were secondary outcomes. On a 1:1 ratio, 46 participants were randomized either to MY (n=23) or PT (n=23). The MY was received once a week for 12 weeks, using a group therapy approach, each session lasting an hour. The PT intervention was sub-categorized further into 6; body and mind (=4), exercise therapy (4), combination of body and mind and exercise therapy (n=8), combination of exercise therapy and pain management (n=2), and ‘no treatment’ (n=5 [n=3 just assessment?]). The PT was delivered to each participant, individually, one to three times a week for ‘several weeks?’, each session lasting 30-45 minutes. Participants were assessed at baseline, 8- and 20-weeks. All treatments were delivered by Physiotherapists. Participants on medications or psychological related treatment continued with their therapies alongside the interventions in this study. Whereas those who were receiving or had received MY and PT interventions within the past 3-months were excluded from participation.

Authors report that at 8-weeks, nearly half of the participants were lost to follow-up. Authors claim that MY is more effective in improving stress, anxiety and HR-QOL than PT interventions. Authors report poor adherence to the intervention as the study’s major limiting factor.

Major issues

1. On page 6, it is not clear what criteria was used to allocate participants to the various conventional PT interventions.

2. The rationale of having other participants not receiving any conventional PT intervention (within the PT as standard care

arm) is not clear

3. ‘Age was considered a confounder, but never met the inclusion criteria for confounders hence the decision to not adjust

the models for age’. Was this, in any way, due to the recruitment (by chance)?

a. Did you consider matching the participants, in both arms, for age, gender and weight? (Potential confounders

for stress, anxiety, and depression [severity/progressiveness/recovery])

i. Could the inability to match the participants limit the interpretation/generalizability of the results?

4. The comparison group (PT) was a heterogeneous. Different participants received different PT conventional interventions

(also with varying dosage?), which might affect the findings differently as well (as reported in your results). This makes the

conclusion that MY is more effective to 'PT' arguable due to lack of standardization of the comparison intervention.

Comparing MY to each sub-group of PT interventions (bearing in mind the sample size calculation approach) would have

been ideal. You may wish to incorporate this in the interpretation and discussion of the results.

Minor issues

1. Line 8 of page 2, consider removing the space between ‘b ut’ (for ‘but’)

2. Line 5 of page 5 , consider replacing ‘control’ with usual/standard care

3. Consider putting SI unit for age in table 1 on page 5

4. ‘Participation during 8-weeks in mean’ column in table 1, is this for number of sessions?

5. Line 5 of page 7 ‘of practical reasons’….. may benefit from a sentence reconstruction

6. Line 2 on page 8 ‘for use with people with…’ may benefit from a sentence reconstruction

7. Line 4 on page 8 ‘to access’, did you want to say ‘to assess’?

8. Lines 8-9 on page 10; the sentence is not clear, may benefit from a sentence reconstruction

9. Lines 8-10 on page 15 may benefit from a sentence reconstruction

10. Lines 18-19 on page 15 may benefit from a sentence reconstruction

a. Do you mean there was no statistical difference between the two groups? Or no statistical changes within groups?

b. The trend of improvement, was it for MY or PT or in both groups?

11. Lines 7-8 on page 16, sentence starting wit ‘consistent with previous research, as in randomized……16’, may benefit from a

sentence reconstruction

Reviewer #3: Summary of short comings

1. Page 1; Write something in the abstract on how data was analysed

2. Page 1: “There was a group trend over time advantaging MediYoga in HADS depression (P = 0.08)”. This sentence is a little confusing. The results were not statistically significant, yet you indicate that there was a trend using that P value.

3. Page 5, line 3: The font color is not consistent with the rest of the text.

4. Page 5, line 5: There is a hyphen as a sign of correction… need to remove it

5. Page 7: Explain how “no assessment and no treatment” is part of PT intervention group

6. Page 7 & 8: indicate the psychometric properties of the instruments used

7. Page 9, paragraph 2: stick to third person instead of first person speech.

8. Page 10-14: Table 4 is too long, crowded and confusing. If possible find a way to separate the results for each outcome instead of combining all outcomes in one table. Of course there will be too many tables but if there is a provision of submitting tables separately, would be good.

9. Is there a reason for presenting the intercept in Table 4? I don’t see the results of intercept being explained in this manuscript.

10. There is some inconsistence in the reference list. For instance for reference 25, there is a need to add other authors.

11. Table 4: it should be “95%CI” and not just “CI”

12. Table 4: Mean should be written together with standard deviation.

13. Page 15, line 4/5: The sentence doesn’t seem complete. HADS had significant difference in comparison to which variable?

14. Page 17, line 2: Majority of the interventions require participants to travel to the clinical site. However, was there something very specific to your study that contributed to the high dropouts?

6. PLOS authors have the option to publish the peer review history of their article (what does this mean?). If published, this will include your full peer review and any attached files.

Reviewer #1: No

Reviewer #2: No

Reviewer #3: No

---

## [Author Response · Author response to Decision Letter 0]

10 Dec 2023

Please see uploaded file response to reviewers.

---

## [Decision Letter · Decision Letter 1]

19 Jan 2024

PONE-D-23-05385R1MediYoga compared to physiotherapy treatment as usual for patients with stress-related symptoms in primary care rehabilitation: a randomized controlled trialPLOS ONE

Dear Dr. Larsson,

Thank you for submitting your manuscript to PLOS ONE. After careful consideration, we feel that it has merit but does not fully meet PLOS ONE’s publication criteria as it currently stands. Therefore, we invite you to submit a revised version of the manuscript that addresses the points raised during the review process.

We look forward to receiving your revised manuscript.

Kind regards,

Enock Madalitso Chisati, PhD

Academic Editor

PLOS ONE

Journal Requirements:

Reviewers' comments:

Reviewer's Responses to Questions

**Comments to the Author**

1. If the authors have adequately addressed your comments raised in a previous round of review and you feel that this manuscript is now acceptable for publication, you may indicate that here to bypass the “Comments to the Author” section, enter your conflict of interest statement in the “Confidential to Editor” section, and submit your "Accept" recommendation.

Reviewer #1: (No Response)

Reviewer #2: All comments have been addressed

2. Is the manuscript technically sound, and do the data support the conclusions?

Reviewer #1: Yes

Reviewer #2: Yes

3. Has the statistical analysis been performed appropriately and rigorously? 

Reviewer #1: Yes

Reviewer #2: Yes

4. Have the authors made all data underlying the findings in their manuscript fully available?

Reviewer #1: Yes

Reviewer #2: Yes

5. Is the manuscript presented in an intelligible fashion and written in standard English?

Reviewer #1: Yes

Reviewer #2: Yes

6. Review Comments to the Author

Reviewer #1: Minor revisions:

1- Line 212: Variance component is not a covariance structure. The three most common underlying covariance structures used in mixed effects linear models are: unstructured, autoregressive, and compound symmetry.

2- Pertaining to the statistical power calculation, within the manuscript indicate that the t-test was the statistical method which attained 80% power. Prior Comment: "Statistical Power Calculation: State the statistical testing method which attains 80% power. Perhaps it is a t-test." And the response: "Yes, the statistical power calculation was done with a t-test."

Reviewer #2: The authors have responded (with scientifically sound justification) to the questions and comments given in the initial review to my satisfactory.

7. PLOS authors have the option to publish the peer review history of their article (what does this mean?). If published, this will include your full peer review and any attached files.

Reviewer #1: No

Reviewer #2: No

---

## [Author Response · Author response to Decision Letter 1]

9 Feb 2024

A rebuttal letter is uploaded with responses to reviewers for the minor revision

---

## [Decision Letter · Decision Letter 2]

5 Mar 2024

MediYoga compared to physiotherapy treatment as usual for patients with stress-related symptoms in primary care rehabilitation: a randomized controlled trial

PONE-D-23-05385R2

Dear Dr. Larsson,

We’re pleased to inform you that your manuscript has been judged scientifically suitable for publication and will be formally accepted for publication once it meets all outstanding technical requirements.

Kind regards,

Enock Madalitso Chisati, PhD

Academic Editor

PLOS ONE

Additional Editor Comments (optional):

Reviewers' comments:

Reviewer's Responses to Questions

**Comments to the Author**

1. If the authors have adequately addressed your comments raised in a previous round of review and you feel that this manuscript is now acceptable for publication, you may indicate that here to bypass the “Comments to the Author” section, enter your conflict of interest statement in the “Confidential to Editor” section, and submit your "Accept" recommendation.

Reviewer #1: (No Response)

2. Is the manuscript technically sound, and do the data support the conclusions?

Reviewer #1: (No Response)

3. Has the statistical analysis been performed appropriately and rigorously? 

Reviewer #1: (No Response)

4. Have the authors made all data underlying the findings in their manuscript fully available?

Reviewer #1: (No Response)

5. Is the manuscript presented in an intelligible fashion and written in standard English?

Reviewer #1: (No Response)

6. Review Comments to the Author

Reviewer #1: (No Response)

7. PLOS authors have the option to publish the peer review history of their article (what does this mean?). If published, this will include your full peer review and any attached files.

Reviewer #1: No

---

## [Editor Report · Acceptance letter]

29 Apr 2024

PONE-D-23-05385R2 

PLOS ONE

Dear Dr. Larsson, 

I'm pleased to inform you that your manuscript has been deemed suitable for publication in PLOS ONE. Congratulations! Your manuscript is now being handed over to our production team.

Kind regards, 

on behalf of

Dr. Enock Madalitso Chisati 

Academic Editor

PLOS ONE